# PRICEFM: FOUNDATION MODEL FOR PROBABILISTIC ELECTRICITY PRICE FORECASTING

## ABSTRACT

Electricity price forecasting in Europe presents unique challenges due to the continent's increasingly integrated and physically interconnected power market. While recent advances in deep learning and foundation models have led to substantial improvements in general time series forecasting, most existing approaches fail to capture the complex spatial interdependencies and uncertainty inherent in electricity markets. In this paper, we address these limitations by introducing a comprehensive and up-to-date dataset across 24 European countries (38 regions), spanning from 2022-01-01 to 2025-01-01. Building on this groundwork, we propose PriceFM, a spatiotemporal foundation model that integrates graph-based inductive biases to capture spatial interdependencies across interconnected electricity markets. The model is designed for multi-region, multi-timestep, and multi-quantile probabilistic electricity price forecasting. Extensive experiments and ablation studies confirm the model's effectiveness, consistently outperforming competitive baselines and highlighting the importance of spatial context in electricity markets.

## 1 INTRODUCTION

The European electricity market is physically interconnected through a network of cross-border transmission lines, enabling the exchange of electricity between regions and optimizing the social welfare at the European level Lago et al. (2018). However, physical constraints, such as limited transmission capacity, can restrict electricity flow between regions and lead to zonal price differences Finck (2021), illustrated in Figure 1. These price disparities highlight the spatial nature of electricity price formation. Recent studies show that electricity price dynamics are strongly influenced by spatial interdependencies and cannot be accurately captured using region-specific models Do et al. (2024). Therefore, explicitly modeling the spatial structure of the European electricity market is essential for producing accurate price forecasts.

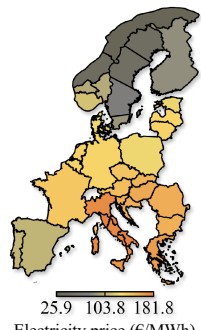 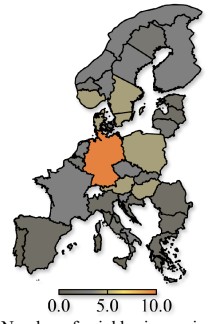

25.9  103.8  181.8
Electricity price (€/MWh)

0.0   5.0   10.0
Number of neighboring regions

Figure 1: Spatial distribution of electricity price and number of neighboring regions. **(a)** Electricity prices for 38 European regions averaged from 2022-01-01 to 2025-01-01. A significant zonal price difference is observed between north and south regions. **(b)** Number of neighboring regions that are *directly* connected to certain region via transmission lines. For example, France (FR) and Portugal (PT) are directly connected to Spain (ES), thus the number of neighboring regions for ES is 2. The mean value across all regions is 3.4.

Most existing studies on electricity price forecasting do not explicitly model the spatial structure and focus on a single-region market, particularly Germany Muniain & Ziel (2020); Maciejowska et al. (2021); Kitsatoglou et al. (2024), as the German market is one of the largest markets in Europe. Other studies explore forecasting methods for markets such as Denmark, Finland, and Spain, also using region-specific models Ziel & Weron (2018); Gianfreda et al. (2020); Loizidis et al. (2024). More recent works explicitly model the spatial nature of the electricity price. For instance, a Graph

Convolutional Network (GCN) is applied to capture spatial interdependencies in the Nordic markets, such as Norway, Sweden, and Finland Yang et al. (2024). Moreover, an attention-based variant is developed to predict prices in certain European markets such as Austria, Germany, and Hungary Meng et al. (2024). However, these models cover only subsets of Europe and primarily produce point forecasts, failing to capture the uncertainty inherent in electricity prices.

Uncertainty modeling in electricity markets is critical, as the electricity price is strongly influenced by intermittent renewable generation and fluctuating demand Lago et al. (2021). Consequently, price forecasting should extend beyond traditional pointwise forecasting to explicitly quantify uncertainties, especially for applications involving risk-sensitive applications such as energy trading and operational planning Ziel & Steinert (2018). A comprehensive survey summarizes various probabilistic forecasting approaches based on quantile regression methods Lago et al. (2021). However, these existing methods often focus on single-region markets, thereby neglecting the rich spatial information.

In recent years, foundation models for time series forecasting have achieved remarkable success across diverse domains, demonstrating strong generalization capabilities by capturing complex data patterns Zhou et al. (2022); Liu et al. (2023); Nie et al. (2023); Wu et al. (2023); Wang et al. (2024), which makes them attractive candidates for electricity price forecasting. In contrast to the conventional notion of foundation models based on pretraining, these models derive their foundation from a generic architecture applicable across domains, and are trained from scratch on the target data Liang et al. (2024). However, existing models are primarily designed for general univariate or multivariate time series tasks and are not explicitly tailored to electricity markets. Moreover, some foundation models only provide pointwise forecasts and thus fail to capture the uncertainty essential for risk-aware trading decisions. Therefore, addressing the spatial interdependencies and uncertainty in electricity price forecasting necessitates a tailored foundation model.

To support the development of foundation models for electricity price forecasting, there is a pressing need for high-quality, large-scale, and up-to-date datasets that reflect the spatiotemporal complexity of integrated European markets. However, existing datasets are often fragmented in structure, cover only short time periods, are outdated, or focus on individual regions Lago et al. (2021). This lack of standardized data poses a significant barrier to training and evaluating foundation models.

In this paper, we introduce a comprehensive and up-to-date dataset and propose PriceFM, a foundation model that incorporates graph-based inductive biases to generate probabilistic forecasts. Similar to other time-series foundation models, PriceFM adopts a generic architecture that can be trained from scratch and applied across various markets. Our contributions are as follows:

**Contribution**

- We introduce and release a comprehensive, up-to-date dataset. To the best of our knowledge, this is the largest and most diverse open dataset for European electricity markets, comprising day-ahead electricity prices, day-ahead forecasts of load, solar, and wind power generation (onshore and offshore), covering 24 European countries (38 regions), spanning from 2022-01-01 to 2025-01-01.

- We propose and release the PriceFM, a novel forecasting framework that integrates prior graph knowledge derived from the spatial topology of the European electricity market. PriceFM supports joint multi-region, multi-timestep, and multi-quantile forecasting.

- We conduct experiments to evaluate the model's performance against multiple baselines, and assess the impact of design choices through ablation studies, thereby providing both quantitative evidence of overall performance and insights into optimal configurations.

## 2 PRELIMINARY

The forecasting target is $\mathcal{T} = 24$ hourly prices for the delivery day $\mathcal{D} + 1$, using data available before gate closure, typically around midday on day $\mathcal{D}$. After midday on $\mathcal{D}$, the electricity prices for $\mathcal{D} + 1$ are published and known. We employ a backward-looking window of size $L$ (e.g. $L = 24$ corresponds to 24 hours from $\mathcal{D}$), for known electricity prices, denoted as $\mathbf{X}_{r_{\text{in}}}^{\text{price}}$. We also include forward-looking exogenous features, such as day-ahead forecasts of load, solar, and wind (onshore and offshore) power generation for $\mathcal{D}+1$, denoted as $\mathbf{X}_{r_{\text{in}}}^{\text{exo}}$, made on $\mathcal{D}$ before gate closure, as well as

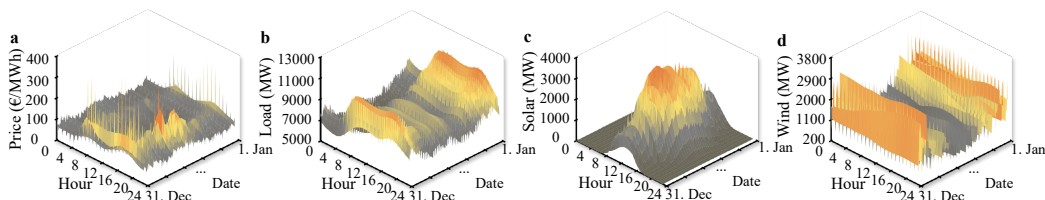

Figure 2: European-level energy data in 2024, averaged across regions. **(a)** Electricity price. Price spikes sharply during the morning and evening peak, dip around midday, and shows higher volatility in the second half of 2024. **(b)** Forecasted load. Load exhibits a double-peak each day with winter peaks substantially larger than summer. **(c)** Forecasted solar power generation. Solar is zero overnight, rises in a smooth bell curve to a strong midday maximum, then falls back to zero by dusk, and is much higher in summer. **(d)** Forecasted wind power generation (onshore and offshore). Wind lacks a daily pattern, fluctuates with high-frequency spikes, and is much higher in winter.

their historical values over $L$. The forecasting setup and the choice of feature set are widely used in prior works Maciejowska (2020); Uniejewski & Weron (2021); Meng et al. (2024). Importantly, this work aims to utilize multi-region inputs to produce multi-region, multi-timestep, and multi-quantile forecasts. Therefore, the input and output are defined as:

- **Input:** $\mathbf{X}_{r_{\mathrm{in}}}^{\mathrm{price}} \in \mathbb{R}^{L \times f_1}$ and $\mathbf{X}_{r_{\mathrm{in}}}^{\mathrm{exo}} \in \mathbb{R}^{(L+\mathcal{T}) \times f_2}$.
- **Output:** $\hat{\mathbf{y}}_{r_{\mathrm{out}}, \tau} \in \mathbb{R}^{\mathcal{T}}$

where $r_{\mathrm{in}}, r_{\mathrm{out}} \in \mathcal{R} = \{\mathrm{AT}, \ldots, \mathrm{SK}\}$ (region codes, detailed in Appendix, Table 1.), $\tau \in \mathcal{Q} = \{0.1, 0.5, 0.9\}$ (quantile levels), $f_1 = 1$, and $f_2$ varies by region.

## 3 DATA

### 3.1 SPATIOTEMPORAL COVERAGE

Spatially, the dataset covers 24 European countries (38 regions). These regions reflect transmission zones rather than administrative boundaries. For example, DK is split into two regions: DK1 and DK2. Each is connected to different regions, resulting in distinct cross-border power flows. Temporally, the dataset spans from 2022-01-01 to 2025-01-01, providing wide temporal coverage.

### 3.2 FEATURE SET

The feature set includes day-ahead electricity prices, load forecasts, and solar and wind power generation forecasts (onshore and offshore). For simplicity, we refer to these features as *price*, *load*, *solar*, and *wind (onshore and offshore)*, respectively. The availability of features across regions is detailed in Appendix, Table 1. A European-level visualization of these features is shown in Figure 2.

### 3.3 RESOLUTION

We resample all features in an hourly resolution, as the raw data exhibit a heterogeneous temporal structure. For example, load from ES is provided at an hourly resolution before 2022-05-23 and then switches to a quarter-hourly resolution afterward; the price from AT is reported hourly, while the load is reported quarter-hourly.

### 3.4 MISSING VALUE

Partial features are excluded due to the high rate (above 15%) of missing values, summarized in Appendix, Table 1. For example, wind offshore from FR has a missing rate of 53.2% and is only available after 2023-08-07; load from SK has a missing rate of 16.8% and is no longer available after 2024-07-01. The features with low missing rates (below 1%) are filled using linear interpolation.

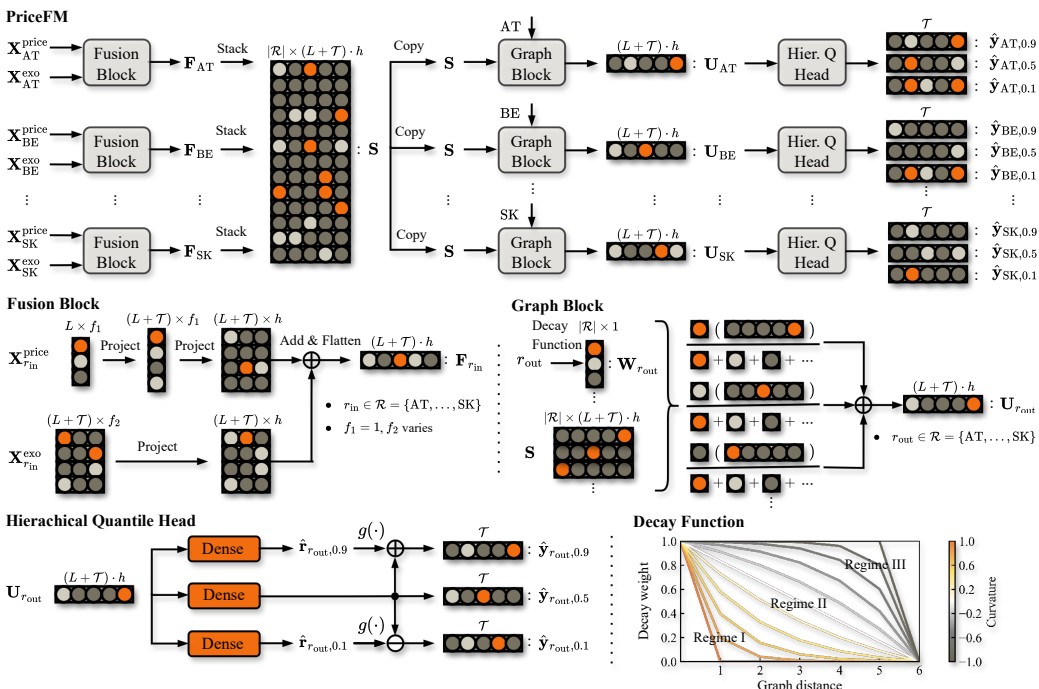

Figure 3: Structure of PriceFM. The input features $\mathbf{X}_{r_{\text{in}}}^{\text{price}}$ and $\mathbf{X}_{r_{\text{in}}}^{\text{exo}}$ are passed into a fusion block to learn regional representation $\mathbf{F}_{r_{\text{in}}}$. These regional representations are then stacked to form the spatial representation $\mathbf{S}$. Next, $\mathbf{S}$ is passed to the graph block to produce the spatial representation $\mathbf{U}_{r_{\text{out}}}$. Finally, $\mathbf{U}_{r_{\text{out}}}$ is fed into hierarchical quantile heads to produce joint forecasts.

## 4 MODEL

PriceFM, illustrated in Figure 3, aims to produce multi-region, multi-timestep, and multi-quantile forecasts. The inputs $\mathbf{X}_{r_{\text{in}}}^{\text{price}} \in \mathbb{R}^{L \times f_1}$ and $\mathbf{X}_{r_{\text{in}}}^{\text{exo}} \in \mathbb{R}^{(L+\mathcal{T}) \times f_2}$ are described in Section *Preliminary*, introducing heterogeneity along the temporal and feature dimensions.

### 4.1 FUSION BLOCK

We first project the temporal dimensions by transposing and projecting $\mathbf{X}_{r_{\text{in}}}^{\text{price}}$ to length $L + \mathcal{T}$ via a dense layer with linear activation, and then transpose back and project the feature dimension of both price and exogenous features into a hidden space of dimension $h$ via $k$ dense layers:

$$\mathbf{X}_{r_{\text{in}}}^{\text{price}} \xrightarrow{\text{Project}} \tilde{\mathbf{X}}_{r_{\text{in}}}^{\text{price}} \in \mathbb{R}^{(L+\mathcal{T}) \times f_1}, \tag{1}$$

$$\tilde{\mathbf{X}}_{r_{\text{in}}}^{\text{price}} \xrightarrow{\text{Project}} \hat{\mathbf{X}}_{r_{\text{in}}}^{\text{price}} \in \mathbb{R}^{(L+\mathcal{T}) \times h}, \tag{2}$$

$$\mathbf{X}_{r_{\text{in}}}^{\text{exo}} \xrightarrow{\text{Project}} \hat{\mathbf{X}}_{r_{\text{in}}}^{\text{exo}} \in \mathbb{R}^{(L+\mathcal{T}) \times h}. \tag{3}$$

Next, we perform feature fusion through residual addition and flatten the fused representation to produce the regional representation $\mathbf{F}_{r_{\text{in}}}$, which encodes the backward-looking price feature and forward-looking contextual information:

$$\mathbf{F}_{r_{\text{in}}} = \text{Flatten}\left(\hat{\mathbf{X}}_{r_{\text{in}}}^{\text{price}} + \hat{\mathbf{X}}_{r_{\text{in}}}^{\text{exo}}\right) \in \mathbb{R}^{(L+\mathcal{T}) \cdot h}. \tag{4}$$

Next, the fused vectors $\mathbf{F}_{r_{\text{in}}}$ for all regions $r_{\text{in}} \in \mathcal{R}$ are stacked to form a spatial representation:

$$\mathbf{S} = \text{Stack}\left(\{\mathbf{F}_{r_{\text{in}}}\}_{r_{\text{in}} \in \mathcal{R}}\right) \in \mathbb{R}^{|\mathcal{R}| \times (L+\mathcal{T}) \cdot h}. \tag{5}$$

## 4.2 GRAPH BLOCK

We first construct *graph distance* by performing a breadth-first search (BFS) traversal on the cross-border grid topology, detailed in Appendix, Table 6. For a given output region $r_{\text{out}} \in \mathcal{R}$, we define the graph distance $d(r_{\text{in}}, r_{\text{out}})$ as the minimal number of transmission hops from each input region $r_{\text{in}}$ to the output region $r_{\text{out}}$, based on direct or indirect physical connectivity:

$$d(r_{\text{in}}, r_{\text{out}}) = \begin{cases} 0 & \text{if } r_{\text{in}} = r_{\text{out}}, \\ 1 & \text{if } r_{\text{in}} \sim r_{\text{out}}, \\ 1 + \min_{r' \sim r_{\text{in}}} d(r', r_{\text{out}}) & \text{otherwise,} \end{cases} \tag{6}$$

where $r_{\text{in}} \sim r_{\text{out}}$ denotes that two regions are directly connected by a transmission line. For example, let $r_{\text{out}} = \text{AT}$. Then $d(\text{AT}, \text{AT}) = 0$. The region HU is directly connected to AT, thus $d(\text{HU}, \text{AT}) = 1$. SK is indirectly connected to AT via HU, yielding $d(\text{SK}, \text{AT}) = 2$.

Next, we introduce a graph decay mechanism to inject prior graph knowledge. Intuitively, input regions that are topologically closer to the output region $r_{\text{out}}$ are expected to exert a stronger influence than more distant ones. To formalize this intuition, we design a *decay function* that modulates the contribution of each neighboring region based on its graph distance, yielding a *decay weight*:

$$w(d; c; D) = \begin{cases} \dfrac{(1 - |c|)^d - (1 - |c|)^D}{1 - (1 - |c|)^D}, & c > 0, \\ 1 - \dfrac{d}{D}, & c = 0, \\ \dfrac{(1 - |c|)^{-d} - (1 - |c|)^{-D}}{1 - (1 - |c|)^{-D}}, & c < 0, \end{cases} \tag{7}$$

where $w(d; c; D) \in [0, 1]$ is the decay weight, $d$ is the graph distance, $c \in [-1, 1]$ is the curvature parameter, and $D$ is the maximum reachable distance from the output region $r_{\text{out}}$. As shown in Figure 3, for $c > 0$ (Regime I), the decay weight drops off sharply with distance, meaning distant regions contribute very little. For $c = 0$ (Regime II), the function reduces to linear decay, decreasing proportionally with graph distance. For $c < 0$ (Regime III), the decay weight decreases more gradually, preserving the influence of neighbors. We construct the *decay mask* as:

$$\mathbf{W}_{r_{\text{out}}} = \begin{bmatrix} w(d(\text{AT}, r_{\text{out}}); c; D) \\ w(d(\text{BE}, r_{\text{out}}); c; D) \\ \vdots \\ w(d(\text{SK}, r_{\text{out}}); c; D) \end{bmatrix} \in \mathbb{R}^{|\mathcal{R}| \times 1}. \tag{8}$$

Next, the learned spatial representation $\mathbf{S}$ is copied $|\mathcal{R}|$ times, each assigned to an output region $r_{\text{out}}$. We inject graph knowledge into $\mathbf{S}$ by computing the decay-weighted average representation:

$$\mathbf{U}_{r_{\text{out}}} = \frac{\mathbf{W}_{r_{\text{out}}}^{\top} \mathbf{S}}{\mathbf{W}_{r_{\text{out}}}^{\top} \mathbf{1}}, \tag{9}$$

where $\mathbf{1} \in \mathbb{R}^{|\mathcal{R}| \times 1}$ is a vector of ones.

This operation acts as spatial regularization and eliminates the need for an exhaustive learning process to determine spatial weights, as required in methods such as the attention mechanism.

## 4.3 HEAD

We design a multi-region, multi-timestep, and multi-quantile head, where the model produces joint probabilistic forecasts. To prevent quantile crossing issue[1], we adopt a hierarchical quantile head Yu et al. (2025). In detail, the median quantile ($\tau = 0.5$) price trajectory, which represents the full set of timesteps $\mathcal{T}$, is predicted from $\mathbf{U}_{r_{\text{out}}}$ via a dense layer:

$$\hat{\mathbf{y}}_{r_{\text{out}}, 0.5} = \text{Dense}(\mathbf{U}_{r_{\text{out}}}) \in \mathbb{R}^{\mathcal{T}}. \tag{10}$$

---

[1]Quantile crossing refers to the phenomenon where upper quantile predictions (e.g., 90%) fall below lower quantiles (e.g., 10%), violating the monotonicity of the quantile function. Chernozhukov et al. (2010).

For the upper quantile ($\tau = 0.9$), a residual price trajectory $\hat{\mathbf{r}}_{r_{\text{out}},0.9}$ is produced from $\mathbf{U}_{r_{\text{out}}}$:

$$\hat{\mathbf{r}}_{r_{\text{out}},0.9} = \text{Dense}(\mathbf{U}_{r_{\text{out}}}) \in \mathbb{R}^{\mathcal{T}}, \tag{11}$$

where a non-negative function $g(\cdot)$, such as absolute-value function, is applied to the price residual. The upper quantile forecast is obtained by adding this non-negative residual to the median:

$$\hat{\mathbf{y}}_{r_{\text{out}},0.9} = \hat{\mathbf{y}}_{r_{\text{out}},0.5} + g(\hat{\mathbf{r}}_{r_{\text{out}},0.9}). \tag{12}$$

For the lower quantile ($\tau = 0.1$), we compute a residual trajectory similarly:

$$\hat{\mathbf{r}}_{r_{\text{out}},0.1} = \text{Dense}(\mathbf{U}_{r_{\text{out}}}) \in \mathbb{R}^{\mathcal{T}}, \tag{13}$$

and subtract it from the median to obtain the lower quantile prediction:

$$\hat{\mathbf{y}}_{r_{\text{out}},0.1} = \hat{\mathbf{y}}_{r_{\text{out}},0.5} - g(\hat{\mathbf{r}}_{r_{\text{out}},0.1}). \tag{14}$$

This hierarchical design guarantees that the upper quantile prediction is greater than or equal to the lower one at each time step, overcoming quantile crossing.

### 4.4 Loss

We introduce the *Average Quantile Loss (AQL)* as the training objective for multi-region, multi-timestep, and multi-quantile probabilistic forecasting. Let $y_{i,r_{\text{out}},t}$ denote the ground-truth price for the $i$-th training sample, output region $r_{\text{out}}$, and timestep $t$, and let $\hat{y}_{i,r_{\text{out}},t,\tau}$ be the corresponding predicted quantile. The AQL is computed as:

$$\text{AQL} = \frac{1}{N\,|\mathcal{R}|\,\mathcal{T}\,|\mathcal{Q}|} \sum_{i=1}^{N} \sum_{r_{\text{out}} \in \mathcal{R}} \sum_{t=1}^{\mathcal{T}} \sum_{\tau \in \mathcal{Q}} L_\tau\left(y_{i,r_{\text{out}},t}, \hat{y}_{i,r_{\text{out}},t,\tau}\right), \tag{15}$$

where $N$ is the number of samples, and the quantile loss $L_\tau$ is defined as:

$$L_\tau(y, \hat{y}_\tau) = \begin{cases} \tau \cdot (y - \hat{y}_\tau), & \text{if } y \geq \hat{y}_\tau, \\ (1 - \tau) \cdot (\hat{y}_\tau - y), & \text{otherwise,} \end{cases} \tag{16}$$

where $y$ and $\hat{y}$ are the true and predicted values, respectively.

## 5 Experiment

We split the data into training (2022-01-01 to 2024-01-01), validation (2024-01-01 to 2024-07-01), and testing (2024-07-01 to 2025-01-01). The choice of the testing period aims to include numerous extreme prices, as illustrated in Figure 2 **(a)**. We assess the model performance using the quantile losses ($Q_{0.1}$, $Q_{0.5}$, and $Q_{0.9}$), AQL, Average Quantile Crossing Rate (AQCR), Root Mean Squared Error (RMSE), Mean Absolute Error (MAE), and Coefficient of Determination ($R^2$). The Diebold-Mariano (DM) test is applied to determine if two models have a significant difference Diebold & Mariano (2002). All metrics are explained in Appendix H. The hyperparameters are detailed in Appendix E

### 5.1 Model Comparison

#### 5.1.1 Naïve Baselines

We include three seasonal naïve baselines as reference models, where only historical prices are used as input: (i) **Naïve**[1] uses 24 prices from the previous day; (ii) **Naïve**[2] uses 24 prices averaged over the past three days; (iii) **Naïve**[3] uses 24 prices averaged over the past seven days. To obtain probabilistic results, we compute empirical quantiles at individual levels ($\mathcal{Q} = \{0.1, 0.5, 0.9\}$) for each delivery hour. The seasonal naïves are commonly used to evaluate the autoregressive strength of the signal and often serve as strong baselines Ziel & Weron (2018); Lago et al. (2021).

The results from Table 1 show that PriceFM significantly outperforms the naïve baselines, confirmed by both the probabilistic and pointwise DM tests, with all $p$-values $< 0.05$ and negative DM values. Specifically, the AQL values of baselines are between 36.83% and 44.80% higher, with an AQCR of 0.00%, as these forecasts are directly computed from historical values. Moreover, the high RMSE and MAE, together with the low $R^2$, observed in the naïve baselines suggest limited performance.

Table 1: Model comparison on the testing set. The symbol "–" indicates that the model does not support probabilistic forecasting by design. All metrics are reported as mean±standard deviation over 5 independent runs. The best results are shown in **bold**, and the second-best are underlined. The units of $Q_{0.1}$, $Q_{0.5}$, $Q_{0.9}$, AQL, RMSE, and MAE are expressed in €/MWh, and AQCR in %.

| Model | $Q_{0.1} \downarrow$ | $Q_{0.5} \downarrow$ | $Q_{0.9} \downarrow$ | AQL $\downarrow$ | AQCR $\downarrow$ | RMSE $\downarrow$ | MAE $\downarrow$ | $R^2 \uparrow$ |
|---|---|---|---|---|---|---|---|---|
| Naïve[1] | 5.68±0.00 | 14.30±0.00 | 8.31±0.00 | 9.43±0.00 | 0.00±0.00 | 46.90±0.00 | 28.60±0.00 | 0.16±0.00 |
| Naïve[2] | 6.01±0.00 | 14.42±0.00 | 8.44±0.00 | 9.62±0.00 | 0.00±0.00 | 46.73±0.00 | 28.84±0.00 | 0.17±0.00 |
| Naïve[3] | 6.65±0.00 | 14.58±0.00 | 8.71±0.00 | 9.98±0.00 | 0.00±0.00 | 46.73±0.00 | 29.15±0.00 | 0.17±0.00 |
| GCN | 4.81±0.16 | 10.76±0.16 | 7.11±0.33 | 7.56±0.15 | 2.19±0.76 | 35.75±0.60 | 21.53±0.33 | 0.51±0.02 |
| GAT | 5.08±0.31 | 11.58±0.37 | 7.90±0.40 | 8.18±0.29 | 1.41±0.47 | 37.63±0.62 | 23.15±0.75 | 0.44±0.01 |
| GraphSAGE | 5.20±0.13 | 11.27±0.23 | 7.39±0.35 | 7.95±0.18 | 2.79±0.56 | 37.09±0.81 | 22.53±0.46 | 0.47±0.02 |
| GraphDiffusion | 4.80±0.20 | 11.03±0.21 | 7.34±0.33 | 7.73±0.18 | 3.33±0.70 | 36.35±0.80 | 22.07±0.42 | 0.48±0.02 |
| GraphARMA | 4.87±0.07 | 11.10±0.22 | 7.00±0.22 | 7.66±0.16 | 2.05±0.65 | 36.15±0.59 | 22.21±0.44 | 0.49±0.02 |
| FEDFormer | – | – | – | – | – | 44.60±0.88 | 27.53±0.81 | 0.30±0.01 |
| PatchTST | – | – | – | – | – | 45.32±1.03 | 26.21±0.92 | 0.29±0.02 |
| iTransformer | – | – | – | – | – | 45.14±0.96 | 27.05±0.64 | 0.29±0.02 |
| TimesNet | – | – | – | – | – | 44.20±0.87 | 26.40±0.52 | 0.30±0.01 |
| TimeXer | – | – | – | – | – | 44.57±0.66 | 26.52±0.55 | 0.30±0.01 |
| **PriceFM** | **4.80±0.06** | **9.81±0.17** | **5.96±0.08** | **6.89±0.12** | **0.00±0.00** | **32.24±0.39** | **19.68±0.31** | **0.61±0.01** |

### 5.1.2 GRAPH MODELS

We compare with multiple GNN variants: (i) **Graph Convolutional Network (GCN)** Kipf (2016), (ii) **Graph Attention Network (GAT)** Veličković et al. (2017), (iii) **GraphSAGE** Hamilton et al. (2017), (iv) **GraphDiffusion** Li et al. (2018), and (v) **GraphARMA** Bianchi et al. (2021). The adjacency matrix of these graph models is explained in Appendix F.

From Table 1, we observe that PriceFM outperforms all graph models, confirmed by both the probabilistic and pointwise DM tests, with all $p$-values $< 0.05$ and negative DM values. Notably, PriceFM reduces the AQL by between 8.86% and 15.77% and consistently achieves 0.00% AQCR compared to graph baselines. A common limitation of these graph baselines is that they lack explicit regularization over noisy regions and rely primarily on data-driven learning to assign spatial importance. As a result, they require large datasets to generalize. Despite our dataset spanning three years, which is considered large in the domain, the daily forecasting requirement limits the training set to only around 700 samples, making it unsuitable for such models.

### 5.1.3 TIME-SERIES FOUNDATION MODELS

We include several time-series foundation models: (i) **FEDFormer** Zhou et al. (2022), (ii) **iTransformer** Liu et al. (2023), (iii) **PatchTST** Nie et al. (2023), (iv) **TimesNet** Wu et al. (2023), and (v) **TimeXer** Wang et al. (2024), to investigate whether these pure time-series models can capture spatial patterns without prior graph knowledge. As these foundation models do not support graph-based input, features from all regions are concatenated along the feature dimension.

The results in Table 1 show that these time-series foundation models achieve similar RMSE and MAE but exhibit better explained variance, as indicated by higher $R^2$, compared to the naïve baselines. Notably, PriceFM outperforms all time-series foundation models, confirmed by the pointwise DM test, with all $p$-values $< 0.05$ and negative DM values. On average, PriceFM improves RMSE, MAE, and $R^2$ by 27.98%, 26.38%, and 0.31, respectively. Given the high complexity of these foundation models, the inclusion of features from noisy regions easily leads to overfitting. This observation confirms that the pure time-series models struggle to recognize useful spatial patterns. In contrast, PriceFM incorporates a graph decay mechanism that acts as a spatial regularizer, thereby attenuating the influence of noisy regions.

Table 2: Ablation studies of different module choices: temporal configuration (rows 1-3), fusion block (rows 4–6), graph block (rows 7–9), and hierarchical quantile head (rows 10–13). The symbol $^\dagger$ marks the method used in PriceFM.

| Method | $Q_{0.1} \downarrow$ | $Q_{0.5} \downarrow$ | $Q_{0.9} \downarrow$ | AQL $\downarrow$ | AQCR $\downarrow$ | RMSE $\downarrow$ | MAE $\downarrow$ | $R^2 \uparrow$ |
|---|---|---|---|---|---|---|---|---|
| $L = 24^\dagger$ | **4.80±0.06** | 9.81±0.17 | 5.96±0.08 | 6.89±0.12 | **0.00±0.00** | **32.24±0.39** | 19.68±0.31 | **0.61±0.01** |
| $L = 72$ | 5.09±0.13 | 10.04±0.20 | 6.27±0.06 | 7.14±0.09 | **0.00±0.00** | 32.76±0.41 | 20.09±0.39 | 0.60±0.01 |
| $L = 168$ | 5.62±0.30 | 10.52±0.26 | 6.65±0.16 | 7.60±0.21 | **0.00±0.00** | 33.82±0.59 | 21.04±0.53 | 0.57±0.01 |
| Res. Add$^\dagger$ | **4.80±0.06** | 9.81±0.17 | 5.96±0.08 | 6.89±0.12 | **0.00±0.00** | **32.24±0.39** | 19.68±0.31 | **0.61±0.01** |
| Concat. | 5.03±0.18 | 10.50±0.21 | 6.40±0.06 | 7.31±0.13 | **0.00±0.00** | 34.68±0.49 | 21.01±0.42 | 0.56±0.01 |
| Cross-Attn | 4.92±0.14 | **9.76±0.10** | 6.07±0.09 | 6.92±0.09 | **0.00±0.00** | 32.46±0.27 | **19.51±0.19** | 0.61±0.01 |
| Decay$^\dagger$ | **4.80±0.06** | 9.81±0.17 | 5.96±0.08 | 6.89±0.12 | **0.00±0.00** | **32.24±0.39** | 19.68±0.31 | **0.61±0.01** |
| Random | 5.32±0.27 | 10.95±0.13 | 7.21±0.11 | 7.83±0.11 | **0.00±0.00** | 35.85±0.30 | 21.89±0.25 | 0.50±0.02 |
| No Decay | 5.23±0.32 | 11.05±0.22 | 6.84±0.22 | 7.71±0.16 | **0.00±0.00** | 35.81±0.63 | 22.11±0.44 | 0.50±0.02 |
| ABS$^\dagger$ | 4.80±0.06 | 9.81±0.17 | 5.96±0.08 | 6.89±0.12 | **0.00±0.00** | 32.24±0.39 | 19.68±0.31 | **0.61±0.01** |
| Square | 4.86±0.14 | **9.80±0.16** | 6.05±0.21 | 6.90±0.15 | **0.00±0.00** | 32.35±0.36 | **19.59±0.33** | 0.61±0.01 |
| ReLU | **4.76±0.15** | 9.84±0.07 | 6.06±0.14 | **6.89±0.07** | **0.00±0.00** | 32.48±0.13 | 19.68±0.15 | 0.61±0.00 |
| Standard | 4.80±0.14 | 10.05±0.06 | **5.94±0.13** | 6.93±0.04 | 4.10±1.28 | 32.76±0.18 | 20.10±0.13 | 0.60±0.01 |

## 5.2 ABLATION STUDY

### 5.2.1 SPATIOTEMPORAL CONFIGURATIONS

- **Curvature Parameter:** Spatially, we evaluate $c \in \{-1.0, -0.8, \ldots, 0.8, 1.0\}$ in increments of 0.2, ranging from weak decay to strong decay. In total, 2,090 trials are conducted to determine the optimal curvature value for each output region individually.

- **Backward-Looking Window Size:** Temporally, we compare $L \in \{24, 72, 168\}$, corresponding to one day, three days, and one week. For each window size, all other hyperparameters are re-optimized.

Spatially, Figure 4 illustrates the testing loss and the distribution of optimal curvature values across all regions. Most regions confirm spatial interdependencies ($c \neq 1.0$). Temporally, the results in Table 2 indicate that the optimal backward-looking window size is 24, potentially because information from the distant past becomes outdated.

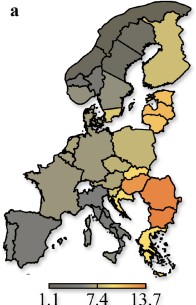
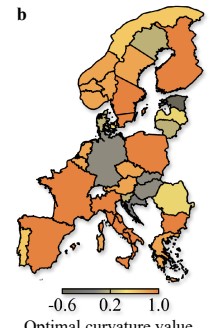

Figure 4: Spatial distribution of testing loss and curvature values. **(a)** Average quantile loss per region on the testing set. Western European regions exhibit lower losses, whereas BG, HU, and RO show particularly high losses (orange areas). **(b)** Optimal curvature value per region. Notably, BG, ES, FR, FI, and partial regions from IT and SE have a curvature value of 1.0, indicating optimal performance by excluding neighboring features.

### 5.2.2 FUSION BLOCK

- **Concatenation:** We replace Equation 4 by first flattening both $\hat{\mathbf{X}}_{r_{\text{in}}}^{\text{price}}$ and $\hat{\mathbf{X}}_{r_{\text{in}}}^{\text{exo}}$, and then concatenating them:

$$\mathbf{F}_{r_{\text{in}}} = \text{Concat}\left(\text{Flatten}\left(\hat{\mathbf{X}}_{r_{\text{in}}}^{\text{price}}\right), \text{Flatten}\left(\hat{\mathbf{X}}_{r_{\text{in}}}^{\text{exo}}\right)\right). \quad (17)$$

- **Cross-Attention:** We apply multi-head attention with $\hat{\mathbf{X}}_{r_{\text{in}}}^{\text{price}}$ as the query and $\hat{\mathbf{X}}_{r_{\text{in}}}^{\text{exo}}$ as both key and value to produce $\hat{\mathbf{X}}_{r_{\text{in}}}^{\text{attn}}$. The attended features are then fused back into the price

representation using residual addition:

$$\mathbf{F}_{r_{\text{in}}} = \text{Flatten}\left(\hat{\mathbf{X}}_{r_{\text{in}}}^{\text{price}} + \hat{\mathbf{X}}_{r_{\text{in}}}^{\text{attn}}\right). \tag{18}$$

The results in Table 2 show that replacing the residual addition with concatenation leads to 6.10% higher AQL for probabilistic prediction. Switching to cross-attention yields comparable performance to residual addition, while introducing additional model parameters. This suggests that the residual addition strikes a favorable balance between predictive performance and model simplicity.

### 5.2.3 GRAPH BLOCK

- **Random Graph Decay Mask:** We replace Equation 8 with a randomly sampled vector, where each decay weight is drawn independently from a uniform distribution over $[0, 1]$, thereby removing the spatial graph prior:

$$\mathbf{W}_{r_{\text{out}}} \sim \mathcal{U}(0, 1)^{|\mathcal{R}| \times 1}. \tag{19}$$

- **No Graph Decay:** We remove the decay mask, which simplifies Equation 9 to a uniform average over input regions:

$$\mathbf{U}_{r_{\text{out}}} = \frac{\mathbf{1}^{\top}\mathbf{S}}{|\mathcal{R}|}, \tag{20}$$

The results in Table 2 demonstrate that randomizing or removing the graph decay mask, which discards the prior graph knowledge, leads to a significant drop in all metrics. We also observe that such results are on par with those of GNN baselines. We emphasize that relying on pure data-driven learning without an explicit decay mechanism leads to a loss of the key inductive bias, limiting the model's performance, especially when the training data is scarce.

### 5.2.4 HIERARCHICAL QUANTILE HEAD

- **Non-Negative Functions:** We replace the absolute-value function used in Equation 12 and 14 with either a square function or ReLU:

$$g(\cdot) = (\cdot)^2, \tag{21}$$

$$g(\cdot) = \max(0, \cdot). \tag{22}$$

- **Standard Multi-Quantile Head:** The Equation 11 and 13 are skipped, and $\mathbf{U}_{r_{\text{out}}}$ is passed directly to three independent dense layers to produce quantile trajectories.

The results in Table 2 reveal that replacing the absolute-value function with either a square function or ReLU does not result in a noticeable change in performance, suggesting that the choice of non-negative function is flexible. Moreover, while the hierarchical quantile head achieves comparable loss to the standard multi-quantile head, the latter exhibits a mean AQCR of 4.10%, indicating that the hierarchical design mitigates quantile crossing without harming performance.

## 6 CONCLUSION

In this paper, we introduced and released a dataset, which will benefit both the research community and the energy industry. We proposed and released PriceFM, a foundation model with a generic architecture applicable to all European electricity markets without relying on pretraining, similar to time-series foundation models used in this work. Extensive experiments and ablation studies demonstrate that PriceFM outperforms competitive baselines and highlight the importance of spatial context. By enabling more accurate and comprehensive probabilistic electricity price forecasting, our work has the potential to support better decision-making in energy trading and grid management.

Several directions remain for future work. First, as the current graph decay function is empirically defined, exploring alternative formulations could improve spatial representations. Second, as the transmission network evolves, model retraining may be required to account for structural changes. Third, since our dataset is sampled at an hourly resolution, we aim to extend PriceFM to support quarter-hourly forecasts, in anticipation of more European bidding zones transitioning to 15-minute markets in the coming years.

**Ethics Statement**   We adhere to the ICLR Code of Ethics. Our study uses market and energy data from the European Network of Transmission System Operators for Electricity (ENTSO-E); no human subjects or personally identifiable information are involved. We release code and documentation for research purposes. We have no known conflicts of interest related to the data providers or outcomes reported.

**Reproducibility Statement**   To support reproducibility, we release well-documented code with an easy-to-use three-step pipeline, along with the code structure and usage guidelines in Appendix B. We report hardware and runtime details in Appendix C to facilitate realistic deployment of our proposed model. As the release and cleaning of the dataset are part of our contributions, the details are described in Section *Data* and Appendix D. Additional reproducibility details are also provided, including data scaling (Appendix G), evaluation metrics (Appendix H), hyperparameters and training procedures (Appendix E).

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

## A  THE USE OF LARGE LANGUAGE MODELS (LLMS)

We employed GPT-4o to assist with grammar correction during the writing process. All LLM-generated suggestions were reviewed and edited to ensure they accurately reflect the authors' original intent. No content related to the methodology, analysis, or reference was generated by the LLM.

## B  CODE GUIDELINE

We open-source all code for preprocessing, modeling, and analysis. The project directory is structured as follows:

```
|- PriceFM/
   |- Data/
   |- Figure/
   |- Model/
   |- Result/
   |- PriceFM.py
   |- Main.py
   |- Tutorial.ipynb
   |- README.md
```

where the `README.md` specifies the required package version. To facilitate reproducibility and accessibility, we have streamlined the entire pipeline through extensive engineering efforts into just three simple steps:

**Step 1:** Create a folder named `PriceFM`, along with subfolders `Data`, `Figure`, `Model`, and `Result`. Place the energy data `EU_Spatiotemporal_Energy_Data.csv` into `Data`, and place `PriceFM.py` inside the `PriceFM` folder.

**Step 2:** Run `Main.py` to process the energy data, and to train, validate, and test the PriceFM. The script `PriceFM.py` contains all necessary functions and classes.

**Step 3:** After execution, you can inspect: `Figure/` for visualizations of forecasts versus true prices; `Model/` for saved model weights; `Result/` for evaluation metrics.

**Optional:** To better understand the code structure and functionality, run `Tutorial.ipynb` block by block.

## C  Hardware and Computation

The PriceFM is evaluated on both an NVIDIA A100 GPU and an Intel Core i7-1265U CPU, respectively. The NVIDIA A100 is designed for high-performance computing and deep learning workloads, offering 80 GB of high-bandwidth memory and up to 6,912 CUDA cores. In contrast, the Intel i7-1265U is a power-efficient CPU commonly found in standard laptops. Under the training setup described in Section *Model*, the training time is approximately 4–5 minutes on the A100 GPU and 12–13 minutes on the i7 CPU. Inference time for both setups is under 10 seconds. We note that neither training nor inference time is critical for our application, as bid submissions can occur at any point before the market gate closure on a daily basis.

## D  Lookup Table and Feature Availability

The country-region code lookup table and the feature availability are listed in Table 1.

## E  Baselines and Hyperparameters

We compare the PriceFM with several spatial and temporal models. The PriceFM is optimized based on validation loss, and the hyperparameter search space is summarized in Table 2. The number of parameters of PriceFM is only 3.38M. We use the Adam optimizer with an initial learning rate of $4 \times 10^{-3}$, which decays exponentially by a factor of 0.95 every 10 epochs. The batch size is 8, which introduces a slight regularization effect. The model is trained for 50 epochs, and the checkpoint with the lowest validation loss is saved.

The graph models, such as GCN, GAT, GraphDiff, GraphSAGE, and GraphARMA, lack an explicit spatial decay mechanism guided by graph distance. GraphConv assigns uniform weights to all neighbors, ignoring spatial relevance; GraphAttn learns attention weights entirely from data without structural priors; GraphDiff uses diffusion kernels that spread information globally, but the importance of nodes is still determined through learned weights, without decay constraints; GraphSAGE aggregates features from sampled neighbors but lacks a notion of spatial proximity; and GraphARMA applies recursive smoothing, which can propagate noise from irrelevant neighbors. As a result, these models require large datasets to recognize spatial patterns and may struggle to suppress the influence of noisy regions. The hyperparameters are optimized based on validation loss, and the search space is summarized in Table 3. All spatial models require an adjacency matrix as input, detailed in Appendix F. Notably, these optimized spatial models contain more than 10M parameters.

FEDFormer, iTransformer, PatchTST, TimesNet, and TimeXer have demonstrated strong performance in general time-series forecasting tasks. FEDFormer, iTransformer, PatchTST, and TimeXer are Transformer-based architectures, while TimesNet is CNN-based. However, these models are not designed for graph forecasting tasks and may require large-scale data to implicitly learn spatial dependencies. The hyperparameters are optimized based on validation loss, and the search space is summarized in Table 4. Notably, the number of parameters of these optimized temporal models ranges from 4.96M to 10.31M, and they are limited to single-region prediction. As a result, 38 separate models must be trained for each output region, since they are not designed to support multi-region forecasting.

## F  Adjacency Matrix

We model the European market as a graph $G = (\mathcal{R}, \mathcal{E})$, where each node $r \in \mathcal{R}$ is a bidding zone and edges indicate direct power flow via cross-border interconnections. This spatial topology is detailed in Table 6. Let $\mathcal{N}(r)$ denote the set of directly connected neighbors of $r$, excluding $r$ itself. The binary adjacency matrix $A \in \{0, 1\}^{|\mathcal{R}| \times |\mathcal{R}|}$ is defined by

$$A_{r,s} = \begin{cases} 1, & \text{if } s \in \mathcal{N}(r), \\ 0, & \text{otherwise,} \end{cases} \quad r, s \in \mathcal{R}. \tag{23}$$

For GNN layers, self-loops can be added via $\tilde{A} = A + I$.

Table 1: Lookup table and feature availability across European regions. ✓ indicates that the feature is available. (✓) denotes partial availability, and the feature is excluded from this study due to the high rate of missing values.

| Country | Region Code | Price | Load | Solar | Wind (Onshore) | Wind (Offshore) |
|---|---|---|---|---|---|---|
| Austria | AT | ✓ | ✓ | ✓ | ✓ | |
| Belgium | BE | ✓ | ✓ | ✓ | ✓ | ✓ |
| Bulgaria | BG | ✓ | ✓ | ✓ | ✓ | |
| Czech Republic | CZ | ✓ | ✓ | ✓ | | |
| Germany, Luxembourg | DE-LU | ✓ | ✓ | ✓ | ✓ | ✓ |
| Denmark | DK1 | ✓ | ✓ | ✓ | ✓ | ✓ |
| Denmark | DK2 | ✓ | ✓ | ✓ | ✓ | ✓ |
| Estonia | EE | ✓ | ✓ | ✓ | ✓ | |
| Spain | ES | ✓ | ✓ | ✓ | ✓ | |
| Finland | FI | ✓ | ✓ | ✓ | ✓ | |
| France | FR | ✓ | ✓ | ✓ | ✓ | (✓) |
| Greece | GR | ✓ | ✓ | ✓ | ✓ | |
| Croatia | HR | ✓ | ✓ | ✓ | ✓ | |
| Hungary | HU | ✓ | ✓ | ✓ | ✓ | |
| Italy | IT-CALA | ✓ | ✓ | ✓ | ✓ | |
| Italy | IT-CNOR | ✓ | ✓ | ✓ | ✓ | |
| Italy | IT-CSUD | ✓ | ✓ | ✓ | ✓ | |
| Italy | IT-NORD | ✓ | ✓ | ✓ | ✓ | |
| Italy | IT-SARD | ✓ | ✓ | ✓ | ✓ | |
| Italy | IT-SICI | ✓ | ✓ | ✓ | ✓ | |
| Italy | IT-SUD | ✓ | ✓ | ✓ | ✓ | |
| Lithuania | LT | ✓ | ✓ | ✓ | ✓ | |
| Latvia | LV | ✓ | ✓ | (✓) | ✓ | |
| Netherlands | NL | ✓ | ✓ | ✓ | ✓ | ✓ |
| Norway | NO1 | ✓ | ✓ | | ✓ | |
| Norway | NO2 | ✓ | ✓ | | ✓ | |
| Norway | NO3 | ✓ | ✓ | | ✓ | |
| Norway | NO4 | ✓ | ✓ | | ✓ | |
| Norway | NO5 | ✓ | ✓ | | | |
| Poland | PL | ✓ | ✓ | ✓ | ✓ | |
| Portugal | PT | ✓ | ✓ | ✓ | ✓ | ✓ |
| Romania | RO | ✓ | ✓ | ✓ | ✓ | |
| Sweden | SE1 | ✓ | ✓ | ✓ | ✓ | |
| Sweden | SE2 | ✓ | ✓ | ✓ | ✓ | |
| Sweden | SE3 | ✓ | ✓ | ✓ | ✓ | |
| Sweden | SE4 | ✓ | ✓ | ✓ | ✓ | |
| Slovenia | SI | ✓ | ✓ | ✓ | | |
| Slovakia | SK | ✓ | (✓) | ✓ | ✓ | |

## G  DATA SCALING

To normalize the data while being robust to extreme values, we employ a `RobustScaler` fitted on the training data, using the `Scikit-Learn` implementation. The fitted scaler is then used to transform validation and testing data.

Table 2: Hyperparameter search space for PriceFM.

| Model | Search Space |
|---|---|
| PriceFM | hidden_size: {12, 24, 48}
layers: {2, 3, 4}
batch_size: {8, 32, 128}
learning_rate: {4e-4, 1e-3, 4e-3}
epochs: 50 |

Table 3: Hyperparameter search space for spatial models.

| Model | Search Space |
|---|---|
| GCN | hidden_size: {32, 128, 512}
layers: {2, 3, 4}
batch_size: {8, 32, 128}
learning_rate: {4e-4, 1e-3, 4e-3}
epochs: 50 |
| GAT | hidden_size: {32, 128, 512}
layers: {2, 3, 4}
n_heads: {2, 4, 8}
batch_size: {8, 32, 128}
dropout: {0.1, 0.3, 0.5}
learning_rate: {4e-4, 1e-3, 4e-3}
epochs: 50 |
| GraphSAGE | hidden_size: {32, 128, 512}
layers: {2, 3, 4}
aggregate: {mean, max, sum}
batch_size: {8, 32, 128}
learning_rate: {4e-4, 1e-3, 4e-3}
epochs: 50 |
| GraphDiff | diff_steps: {2, 4, 6}
hidden_size: {32, 128, 512}
layers: {2, 3, 4}
batch_size: {8, 32, 128}
learning_rate: {4e-4, 1e-3, 4e-3}
epochs: 50 |
| GraphARMA | hidden_size: {32, 128, 512}
layers: {2, 3, 4}
order: {1, 2, 4}
iteration: {1, 2, 4}
batch_size: {8, 32, 128}
learning_rate: {4e-4, 1e-3, 4e-3}
epochs: 50 |

# H  METRICS

## H.1  QUANTILE LOSS AT INDIVIDUAL LEVELS

We compute quantile loss separately for each target quantile:

$$Q_\tau = \frac{1}{N\,|\mathcal{R}|\,\mathcal{T}} \sum_{i=1}^{N} \sum_{r_{\text{out}} \in \mathcal{R}} \sum_{t=1}^{\mathcal{T}} L_\tau \left( y_{i,r_{\text{out}},t}, \hat{y}_{i,r_{\text{out}},t,\tau} \right), \tag{24}$$

Table 4: Hyperparameter search space for time-series foundation models.

| Model | Search Space |
|---|---|
| FEDFormer | hidden_size: {32, 128, 512}
conv_hidden_size: {32, 128, 512}
e_layers: {2, 3, 4}
n_heads: {2, 4, 8}
dropout: {0.1, 0.3, 0.5}
batch_size: {8, 32, 128}
learning_rate: {4e-4, 1e-3, 4e-3}
epochs: 50 |
| iTransformer | hidden_size: {32, 128, 512}
e_layers: {2, 3, 4}
d_ff: {512, 1024, 2048}
n_heads: {2, 4, 8}
dropout: {0.1, 0.3, 0.5}
batch_size: {8, 32, 128}
learning_rate: {4e-4, 1e-3, 4e-3}
epochs: 50 |
| PatchTST | hidden_size: {32, 128, 512}
e_layers: {2, 3, 4}
n_heads: {2, 4, 8}
dropout: {0.1, 0.3, 0.5}
patch_len: {4, 6, 12}
batch_size: {8, 32, 128}
learning_rate: {4e-4, 1e-3, 4e-3}
epochs: 50 |
| TimesNet | hidden_size: {32, 128, 512}
conv_hidden_size: {32, 128, 512}
e_layers: {2, 3, 4}
dropout: {0.1, 0.3, 0.5}
batch_size: {8, 32, 128}
learning_rate: {4e-4, 1e-3, 4e-3}
Epochs: 50 |
| TimeXer | hidden_size: {32, 128, 512}
e_layers: {2, 3, 4}
n_heads: {2, 4, 8}
d_ff: {512, 1024, 2048}
dropout: {0.1, 0.3, 0.5}
batch_size: {8, 32, 128}
learning_rate: {4e-4, 1e-3, 4e-3}
epochs: 50 |

where $\tau \in \{0.1, 0.5, 0.9\}$.

## H.2 AVERAGE QUANTILE LOSS (AQL)

AQL represents the average quantile loss across all quantiles, as described in Section *Model*.

## H.3 AVERAGE QUANTILE CROSSING RATE (AQCR)

AQCR captures the proportion of forecasted distributions that violate quantile monotonicity, i.e., when a lower quantile is predicted to be greater than a higher one. For each sample, the quantile

crossing indicator is defined as:

$$C_{i,r,t} = \mathbb{I}\left(\max_{\tau_l < \tau_u} \left(\hat{y}_{i,r,t,\tau_l} - \hat{y}_{i,r,t,\tau_u}\right) > 0\right) \tag{25}$$

where $\mathbb{I}(\cdot)$ is an indicator function that returns 1 if any quantile pair fulfills the condition inside and 0 otherwise.

We compute the AQCR as:

$$\text{AQCR} = \frac{1}{N\,|\mathcal{R}|\,\mathcal{T}} \sum_{i=1}^{N} \sum_{r \in \mathcal{R}} \sum_{t=1}^{\mathcal{T}} C_{i,r,t}. \tag{26}$$

A lower AQCR indicates fewer quantile crossing violations and thus reflects more reliable probabilistic forecasts.

### H.4 ROOT MEAN SQUARED ERROR (RMSE)

We compute RMSE within each region, then average over all regions:

$$\text{RMSE}_r = \sqrt{\frac{1}{N\mathcal{T}} \sum_{i=1}^{N} \sum_{t=1}^{\mathcal{T}} \left(y_{i,r,t} - \hat{y}_{i,r,t,0.5}\right)^2}, \tag{27}$$

$$\text{RMSE} = \frac{1}{|\mathcal{R}|} \sum_{r \in \mathcal{R}} \text{RMSE}_r. \tag{28}$$

### H.5 MEAN ABSOLUTE ERROR (MAE)

Same procedure as RMSE, but using absolute error:

$$\text{MAE}_r = \frac{1}{N\mathcal{T}} \sum_{i=1}^{N} \sum_{t=1}^{\mathcal{T}} \left|y_{i,r,t} - \hat{y}_{i,r,t,0.5}\right|, \tag{29}$$

$$\text{MAE} = \frac{1}{|\mathcal{R}|} \sum_{r \in \mathcal{R}} \text{MAE}_r. \tag{30}$$

### H.6 COEFFICIENT OF DETERMINATION

We compute the Coefficient of Determination ($R^2$) for each region and average across all regions:

$$\text{R}_r^2 = 1 - \frac{\sum_{i=1}^{N} \sum_{t=1}^{\mathcal{T}} \left(y_{i,r,t} - \hat{y}_{i,r,t,0.5}\right)^2}{\sum_{i=1}^{N} \sum_{t=1}^{\mathcal{T}} \left(y_{i,r,t} - \bar{y}_r\right)^2}, \tag{31}$$

$$\bar{y}_r = \frac{1}{N\mathcal{T}} \sum_{i=1}^{N} \sum_{t=1}^{\mathcal{T}} y_{i,r,t}, \tag{32}$$

$$\text{R}^2 = \frac{1}{|\mathcal{R}|} \sum_{r \in \mathcal{R}} \text{R}_r^2. \tag{33}$$

### H.7 DIEBOLD & MARIANO (DM) TEST

To assess whether differences in forecasting performance are statistically significant, we apply the DM test. We compute the loss differential at each prediction instance.

For probabilistic forecasts, we compute the loss differential at each quantile $\tau \in \mathcal{Q}$ between two models $l \in \{1, 2\}$:

$$d_{i,r,t,\tau} = L_\tau\left(y_{i,r,t}, \hat{y}_{i,r,t,\tau}^{(1)}\right) - L_\tau\left(y_{i,r,t}, \hat{y}_{i,r,t,\tau}^{(2)}\right). \tag{34}$$

For point forecasts, the loss differential between two models is computed for each sample, region, and timestep as:

$$d_{i,r,t} = \left| y_{i,r,t} - \hat{y}_{i,r,t}^{(1)} \right| - \left| y_{i,r,t} - \hat{y}_{i,r,t}^{(2)} \right|. \tag{35}$$

The DM test statistic is then calculated as:

$$\mathrm{DM} = \frac{\bar{d}}{\hat{\sigma}_d / \sqrt{M}}, \tag{36}$$

$$\bar{d} = \frac{1}{M} \sum_{j=1}^{M} d_j, \tag{37}$$

where $M = N \cdot |\mathcal{R}| \cdot \mathcal{T} \cdot |\mathcal{Q}|$ for probabilistic forecasts, and $M = N \cdot |\mathcal{R}| \cdot \mathcal{T}$ for point forecasts. The index $j$ enumerates all prediction instances across dimensions, and $\hat{\sigma}_d$ is the sample standard deviation of $\{d_j\}_{j=1}^{M}$. We compute a $p$-value; if $p < 0.05$ and the DM value is positive (negative), then we report that model 2 (model 1) significantly outperforms the other in Section *Experiment*. The rules are summarized in Table 5.

Table 5: Interpretation of DM test outcomes.

| Condition | Interpretation | Conclusion |
|---|---|---|
| $p < 0.05$, DM $> 0$ | Statistically significant | Model 2 is better |
| $p < 0.05$, DM $< 0$ | Statistically significant | Model 1 is better |
| $p \geq 0.05$ | Not statistically significant | – |

Table 6: Direct neighbors by region and neighbor count.

| Region Code | Direct Neighbors | Count |
|---|---|---|
| AT | CZ, DE-LU, HU, IT-NORD, SI | 5 |
| BE | DE-LU, FR, NL | 3 |
| BG | GR, RO | 2 |
| CZ | AT, DE-LU, PL, SK | 4 |
| DE-LU | AT, BE, CZ, DK1, DK2, FR, NL, NO2, PL, SE4 | 10 |
| DK1 | DE-LU, DK2, NL, NO2, SE3 | 5 |
| DK2 | DE-LU, DK1, SE4 | 3 |
| EE | FI, LV | 2 |
| ES | FR, PT | 2 |
| FI | EE, NO4, SE1, SE3 | 4 |
| FR | BE, DE-LU, ES, IT-NORD | 4 |
| GR | BG, IT-SUD | 2 |
| HR | HU, SI | 2 |
| HU | AT, HR, RO, SI, SK | 5 |
| IT-CALA | IT-SICI, IT-SUD | 2 |
| IT-CNOR | IT-CSUD, IT-NORD | 2 |
| IT-CSUD | IT-CNOR, IT-SARD, IT-SUD | 3 |
| IT-NORD | AT, FR, IT-CNOR, SI | 4 |
| IT-SARD | IT-CSUD | 1 |
| IT-SICI | IT-CALA | 1 |
| IT-SUD | GR, IT-CALA, IT-CSUD | 3 |
| LT | LV, PL, SE4 | 3 |
| LV | EE, LT | 2 |
| NL | BE, DK1, DE-LU, NO2 | 4 |
| NO1 | NO2, NO3, NO5, SE3 | 4 |
| NO2 | DE-LU, DK1, NL, NO1, NO5 | 5 |
| NO3 | NO1, NO4, NO5, SE2 | 4 |
| NO4 | FI, NO3, SE1, SE2 | 4 |
| NO5 | NO1, NO2, NO3 | 3 |
| PL | CZ, DE-LU, LT, SE4, SK | 5 |
| PT | ES | 1 |
| RO | BG, HU | 2 |
| SE1 | FI, NO4, SE2 | 3 |
| SE2 | NO3, NO4, SE1, SE3 | 4 |
| SE3 | DK1, FI, NO1, SE2, SE4 | 5 |
| SE4 | DE-LU, DK2, LT, PL, SE3 | 5 |
| SI | AT, HR, HU, IT-NORD | 4 |
| SK | CZ, HU, PL | 3 |

