# OpenReview forum: "PriceFM: Foundation Model for Probabilistic Electricity Price Forecasting"
_ICLR.cc/2026/Conference — ICLR 2026 Conference Withdrawn Submission_

### Official Review · Reviewer_oUog · 2025-10-19

**Soundness:** 3
**Presentation:** 4
**Contribution:** 2
**Rating:** 6
**Confidence:** 4

**Summary:**

This paper presents PriceFM, a foundation model for probabilistic forecasting of electricity prices in the European market. The model integrates spatiotemporal dependencies by combining price data, exogenous variables (e.g., load, solar, and wind forecasts), and cross-regional connectivity information to predict day-ahead electricity prices across 24 hours. PriceFM is trained and evaluated on a newly compiled, large-scale dataset covering 38 European regions between 2022 and 2025.

**Strengths:**

- **High-impact and socially relevant problem**: Electricity price forecasting is a critical and high-value task with implications for grid stability, market regulation, and renewable integration. Addressing it effectively could benefit a wide range of stakeholders across the energy ecosystem.

- **Large-scale and comprehensive analysis**: The study is conducted at an unprecedented European scale, integrating data from multiple interconnected markets. To the best of my knowledge, this is among the first works to tackle electricity price forecasting at continental resolution, which greatly enhances its practical relevance.

- **Strong empirical performance**: PriceFM achieves consistently positive results across benchmarks, outperforming or matching strong baselines in most regions and time horizons.

- **Extensive ablation study**: The paper includes a thorough and well-designed ablation analysis, providing clear insights into the contribution of each architectural component and validating the robustness of the proposed approach.

**Weaknesses:**

- **Limited exploration of foundation model capabilities**:One of the defining aspects of a foundation model is its ability to generalize and adapt to downstream tasks via fine-tuning or task-specific adaptation. While the paper presents strong results for day-ahead price forecasting, it does not explore whether PriceFM retains its performance when adapted to other related forecasting problems, such as load, renewable generation, or multi-step prediction. Demonstrating this adaptability would strengthen the claim of PriceFM being a true foundation model.

- **Missing simple baselines for context**: It is well known that simple models such as XGBoost can achieve competitive results in electricity price forecasting. Even if these models might not outperform deep architectures at large scales, including them would provide valuable context for understanding how much PriceFM improves over widely adopted, industry-standard approaches.

- **Clarity of exposition and figures**: Although the method is understandable, some parts of the exposition could be clearer. Figure 3 and the explanations in Section 4 are somewhat difficult to follow. A higher-level schematic in Section 3 showing the overall architecture, complemented by more focused sub-figures or diagrams for the specific modules in Section 4, could improve readability. However, I understand that space limitations might make this challenging.

- **Discussion on temporal window length**: The paper states that the optimal input window size is 24 hours, which is somewhat surprising given prior work showing that longer temporal lags can be beneficial [1], especially for exogenous variables that encode past consumption and generation patterns. Could the authors elaborate on this? Is this limitation due to the architecture (e.g., difficulty modeling long-term dependencies), or has it another reason? Extending the model with transformer-based encoders might allow the incorporation of longer temporal contexts. This does not undermine the contribution, but a deeper discussion would help delineate the model’s limitations and potential future extensions, as we are talking about a foundation model.

- **Training data representativeness**: The main concern lies in the training period. The training dataset spans 2022–2024, a period heavily affected by the European energy crisis following the war in Ukraine, which led to reduced Russian gas supply, significant price spikes, and direct market interventions by national governments [2–4]. As a result, the model might have learned representations tied to an anomalous regime rather than general market dynamics. Since this work positions PriceFM as a foundation model, it would be important to test its robustness and transferability to other periods, for example, by training on pre-crisis data, performing zero-shot evaluation, or exploring simple fine-tuning on different years to assess its adaptability under more typical market conditions.

[1] Vega-Márquez, B., Solís-García, J., Nepomuceno-Chamorro, I. A., & Rubio-Escudero, C. (2024). A comparison of time series lags and non-lags in Spanish electricity price forecasting using data science models. Logic Journal of the IGPL, 32(6), 1036-1047.
[2] https://www.ecb.europa.eu/press/economic-bulletin/focus/2022/html/ecb.ebbox202204_01~68ef3c3dc6.en.html
[3] https://www.euronews.com/business/2025/02/24/three-years-on-how-russias-invasion-reshaped-energy-prices-across-europe
[4] https://www.reuters.com/world/europe/hold-eus-solidarity-with-ukraine-unshakeable-commission-chief-says-2022-09-14/

**Questions:**

- **Model capacity**: Could the authors provide details on the number of parameters in PriceFM (and optionally compare it to the strongest baselines)? This information would help assess the model’s complexity, scalability, and computational footprint relative to existing approaches.

- **Addressing reviewer concerns**: I would be glad to revise my evaluation if the authors can improve upon some of the points mentioned above, particularly regarding the model’s generalization to other time periods, its potential for downstream adaptation or fine-tuning, and the discussion on temporal window length.

---

### Official Review · Reviewer_GeN4 · 2025-10-29

**Soundness:** 2
**Presentation:** 3
**Contribution:** 2
**Rating:** 2
**Confidence:** 3

**Summary:**

This paper aims to tackle an important application – electricity price forecast - where spatial interdependencies and uncertainty are essential. It contributes a dataset of the European electricity market and PriceFM with a spatial decay prior. Compared to a set of baselines, numerical results demonstrate that PriceFM is effective. However, there are technical issues concerning its positioning as a foundation model, novelty justification of methodological design and numerical evidence of PriceFM’s robustness and generalizability.

**Strengths:**

- This paper identifies a real-world need for high-quality, large-scale, and up-to-date datasets of electricity prices and release a comprehensive datasets of electricity market.
- This paper proposes a foundation model called PriceFM for price forecasting that incorporates spatial topology of the European electricity market.
- A set of baselines are compared with the proposed PriceFM to validate the effectiveness of PriceFM.

**Weaknesses:**

- The proposed PriceFM model adopts several building blocks, such as fusion block, graph block, and a versatile head, which seem to be functioning. But the design in Section 4 lacks proper novelty justification compared to the SOTA, including those baselines considered.
- The graph decay mechanism is empirically designed and lacks grounding in the fundamental principles of electricity markets. Electricity markets across different countries and regions are interconnected through power networks, but the current mechanism does not seem to capture underlying fundamentals.
- In the forecasting setup, the backward-looking window include known prices, but it seems that it does not mention other exogenous features are included in the backward-looking window as well. For the forward-looking window features, such as day-ahead forecasts of load and renewable, how do the forecast quality affect the performance needs more verification and discussion.
- The electricity market is experiencing a fundamental transformation to net zero with more renewables and responsive demand. How would the evolving changes in the electricity market affect the effectiveness of PriceFM, say more recently in 2025? More study and discussions will be needed to demonstrate the model robustness.
- The developed foundation model looks task-specific with a strong emphasis on spatial interdependencies with case study in the European electricity market, which may raise questions of whether it has a bias and cannot be generalized. It’d be good to have more evidence and discussion.
- A minor comment: the electricity market data are usually publicly available and anyone who spent some time (not too long) could compile the data. Releasing the dataset in this work is claimed as the first contribution, which raise questions about what additional value or novelty the authors provide beyond data collection.

**Questions:**

See the weaknesses.

---

### Official Review · Reviewer_WggC · 2025-10-31

**Soundness:** 3
**Presentation:** 3
**Contribution:** 2
**Rating:** 4
**Confidence:** 4

**Summary:**

This paper introduces PriceFM, a foundation model for probabilistic electricity price forecasting across 24 European countries (38 regions) over 2022–2025. The authors construct a comprehensive spatiotemporal dataset integrating electricity price, load, solar, and wind forecasts, and design a model that embeds graph-based inductive biases to capture spatial interdependencies among interconnected electricity markets. The model jointly predicts multi-region, multi-timestep, and multi-quantile forecasts, employing a hierarchical quantile head to ensure monotonicity among quantiles. Extensive experiments benchmark PriceFM against naïve seasonal baselines, graph neural networks (GCN, GAT, GraphSAGE, GraphDiffusion, GraphARMA), and several time-series foundation models (FEDFormer, PatchTST, iTransformer, TimesNet, TimeXer), showing PriceFM’s superior probabilistic and deterministic forecasting performance.

**Strengths:**

(1) Releases the most comprehensive open dataset for European electricity markets to date, covering 24 countries and 38 regions over three years, enabling continent-scale modeling and establishing a valuable benchmark for future research;
(2) The authors design a unified framework that jointly handles multi-region, multi-timestep, and multi-quantile probabilistic forecasting within an end-to-end architecture, demonstrating strong practical applicability;
(3) The proposed graph decay mechanism effectively embeds physical grid topology as an inductive bias, enhancing spatial interpretability and maintaining robustness under limited data conditions. Comprehensive ablation studies further validate the necessity and effectiveness of key design choices, such as the decay mechanism and residual fusion;
(4) The authors’ commitment to open-source code, data, and training pipelines ensures reproducibility and facilitates subsequent comparative research within the community.

**Weaknesses:**

(1) The theoretical foundation of the proposed graph decay function is insufficient. It relies on a fixed distance-based weighting scheme without learnable parameters, and the paper lacks theoretical or empirical evidence to justify this specific formulation.
(2) The model operates only at an hourly resolution, with no validation on finer temporal scales such as 15-minute intervals, limiting its practical applicability to modern electricity markets.
(3) The experimental scope is incomplete. Comparisons with recent advanced spatiotemporal GNNs and domain-specific probabilistic models are missing, making it difficult to assess the method’s relative competitiveness.
(4) Labeling PriceFM as a “foundation model” may be overstated, as it is trained from scratch on a single-domain dataset and does not demonstrate cross-task or cross-domain generalization.
(5) The authors should compare the relevent papers such as https://doi.org/10.1016/j.inffus.2024.102836.
(6) The paper does not explicitly analyze model performance under extreme price events (e.g., price spikes), which are particularly critical for market risk management and forecasting robustness.

**Questions:**

(1) The paper does not provide direct evidence of model interpretability or feature attribution analysis. Could the authors include such an analysis to better understand which features or regions contribute most to PriceFM’s predictions?
(2) Could you please further clarify why PriceFM is described as a “foundation model”? Is this classification based mainly on its general-purpose architecture, rather than on pre-training or demonstrated cross-domain generalization capabilities?
(3) Compared with learnable attention mechanisms, in what scenarios do you believe the current fixed, prior-based graph decay mechanism has distinct advantages, and what are its potential limitations in capturing dynamic spatial dependencies?
(4) Regarding the experimental evaluation, have you considered comparing PriceFM with more specialized spatiotemporal forecasting models or domain-specific probabilistic models for electricity markets to better contextualize its competitiveness within the field?
(5) The analytical form of the decay function (Equation 7) is intriguing. Could you elaborate on whether it has any deeper physical or mathematical rationale, or whether it was primarily designed through empirical observation and performance tuning?

---

### Official Review · Reviewer_V8m3 · 2025-10-31

**Soundness:** 3
**Presentation:** 3
**Contribution:** 2
**Rating:** 4
**Confidence:** 4

**Summary:**

This paper introduces PriceFM, a spatiotemporal model for multi‑region, multi‑horizon, multi‑quantile day‑ahead electricity price forecasting in Europe. The authors (i) curate an open dataset covering 24 countries / 38 bidding zones from 2022‑01‑01 to 2025‑01‑01 with prices and day‑ahead forecasts for load, solar, and wind (on/offshore), (ii) propose a simple architecture that fuses regional features, injects graph‑distance priors via a decay mask computed from cross‑border interconnections, and outputs monotone quantiles using a hierarchical quantile head, and (iii) report improvements over seasonal naives, common GNN variants, and several time‑series "foundation models,"
with statistically significant gains on RMSE/MAE and lower quantile losses.

**Strengths:**

I like the paper's novelty: it combines a hard, topology‑aware inductive bias, which in the paper is a parametric graph‑distance decay with curvature, also with a lightweight fusion head for probabilistic forecasting. Compared to fully learned attention/GNNs, this is a clear design stance: inject physics‑inspired priors to fight overfitting under data scarcity (around 700 daily samples). The hierarchical quantile head enforces monotonicity and yields AQCR = 0% without degrading AQL This was a solid design choice.  The dataset the authors provide is also very likely useful to this community.

The paper contains thorough ablations and comparisons against other state-of-the art methods.

**Weaknesses:**

The "Foundation model" framing is overstated. The model is trained from scratch on one domain and does not demonstrate cross‑task/region transfer, pretraining, or in‑context generalization typically associated with "foundation models." The paper partially reframes "foundation" as reusable architecture (citing a survey), but experiments do not evaluate zero‑shot to unseen regions, fine‑tuning efficiency, or few‑shot adaptation that would substantiate the term. Consider tempering the claim or adding transfer studies.

There are some baseline fairness / inconsistencies.

Quantile‑capable baselines are missing. For probabilistic forecasting, strong references like TFT (Temporal Fusion Transformer) with quantile loss, MQ‑RNN/DeepAR‑style distributional heads, LightGBM‑QR, or NGBoost are standard: current TS FMs are evaluated only on point metrics (no AQL/CRPS), limiting the strength of the probabilistic claim.

There is an internal inconsistency: Sec. 5.1.3 says TS FMs ingest all regions concatenated (single multi‑region model), but Appendix E states they are "limited to single‑region prediction" requiring 38 separate models. Which setting produced Table 1? This affects both capacity and data exposure comparisons. Please clarify.

The statistical testing details are insufficient. The Diebold‑Mariano (DM) test is computed by pooling all instances (Appendix H.7), but for multi‑step daily horizons the loss differential is likely serially correlated. No HAC/Newey‑West variance correction is mentioned, which may inflate significance. Clarify whether a robust DM variant was used.

The calibration is not evaluated. Beyond AQL and AQCR, probabilistic quality should include coverage, interval width, PIT histograms, or CRPS/WIS. This is crucial in markets with fat tails and spikes (the chosen test period intentionally contains many). Without calibration evidence, the practical value for risk metrics (VaR/ES) is unclear.

The "Joint" forecasting claim is ambiguous. The head outputs quantiles across all T simultaneously, but the paper does not model or evaluate joint dependence across time or regions (e.g., multivariate calibration or energy score). Calling the output "joint" may be misleading, currently it is vector‑valued marginal quantiles with within‑vector monotonicity.

Overall, the paper has some solid points, but there are many circumstances that show limited novelty in terms of the paper's real-world applicability. For example, the graph prior is static and overly coarse: the decay relies on topological hop distance computed via BFS and a scalar curvature per region. European interzonal congestion, NTCs, and physical flows are highly time‑varying; a static, unweighted distance likely mis‑specifies influence during islanding or congestion events. We need to see how good the results are during congestion events.

**Questions:**

For FEDformer / iTransformer / PatchTST / TimesNet / TimeXer, did you train one multi‑region model (features concatenated, as in Sec. 5.1.3) or 38 region‑specific models (as in Appendix E)? Please align the paper and specify parameter counts and data exposure accordingly for Table 1.

Why were TFT with quantile loss, MQ‑RNN/DeepAR (distributional), or tree‑based quantile regressors omitted? Would you consider adding at least TFT‑Quantile and reporting AQL/CRPS/coverage?

Did you apply HAC/Newey‑West correction for multi‑step serial correlation? If not, please recompute DM statistics or add caveats.

---

### Note · Authors · 2025-12-03

I have read and agree with the venue's withdrawal policy on behalf of myself and my co-authors.